# Protocol: Weight-adjusted effective volume of 0.5% ropivacaine for combined costoclavicular brachial plexus block–cervical plexus blocks undergoing arthroscopic shoulder surgery: A dose-finding study protocol

Jianlin Wang[1], Liyong Yuan[1], Zhong Mei[2], Zhimin Sheng[3], Xiaolu Huang[4], Miao Zhu[1]*

1 Department of Anesthesiology, Ningbo No.6 Hospital, Ningbo University School of Medicine, Ningbo, Zhejiang, China, 2 Department of Anesthesiology, Zhejiang Xiaoshan Hospital, Hangzhou, China, 3 Department of Anesthesiology, Wenling Maternity and Child Health Care Hospital, Taizhou, China, 4 Department of Operating Room, Yinzhou No. 2 Hospital, Ningbo, Zhejiang, China

* zhumiao002@163.com

## Abstract

### Introduction

Rotator cuff injuries are common clinically, and arthroscopic repair is widely applied. Postoperative analgesia can be assisted by the interscalene brachial plexus block; however, it comes with side effects, among which a high incidence of hemidiaphragmatic paralysis (HDP) is included. Costoclavicular brachial plexus–cervical plexus blocks (CCB–CPBs) offer comparable analgesia with lower HDP risk, yet local anesthetic volume issues can affect outcomes. In patients undergoing arthroscopic shoulder surgeries under general anesthesia with CCB–CPBs, the aim is to determine the optimal dose of ropivacaine for postoperative analgesia while avoiding hemidiaphragmatic paralysis (HDP),

### Methods and analysis

This trial will be a prospective, single-arm, double-blind dose finding study. We plan to enroll 40 patients who will be scheduled to undergo arthroscopic shoulder surgeries under anesthesia that combines general anesthesia with CCB–CPBs. The volume of the local anesthetic will be determined by adopting the Up-and-Down sequential allocation study design. The primary outcome will be the numerical rating scale (NRS) scores of the patients prior to their departure from the post-anesthesia care unit (PACU). As for the secondary outcomes, they will include the ipsilateral diaphragmatic excursion, the characteristics of the sensory–motor block, the occurrence of complications, as well as the consumption of fentanyl during the operation.

**Data availability statement:** No datasets were generated or analysed during the current study. All relevant data from this study will be made available upon study completion.

**Funding:** Agricultural and Social Development Project in Yinzhou, Ningbo, Zhejiang (No. 2024AS063) The peer review comments of the fund have not been made public. Therefore, we are unable to provide the specific peer review opinions.

**Competing interests:** The authors have declared that no competing interests exist.

## Ethics and dissemination

Approval for the protocol of this study was granted by the Ethics Committee of Ningbo No. 6 Hospital in Zhejiang Province, China, on July 29, 2024 (Approval No. 2024-67L). Once the study is completed, we are committed to guaranteeing that the results will be accessible to the public, irrespective of the outcome. This will involve either publishing them in an appropriate journal or presenting them orally at academic conferences.

## Trial registration

Trial registration number ChiCTR2400090292

## Introduction

Rotator cuff injuries are rather prevalent in clinical practice. The overall prevalence rate escalates with age, approximately steadily increasing from approximately 9.7% to approximately 62% [1]. Presently, arthroscopic rotator cuff repair is being applied more and more widely in clinical settings. Data sourced from the UK show that the quantity of individuals who underwent rotator cuff repair and subacromial decompression increased tenfold from 2004 to 2010 [2].

The interscalene brachial plexus block (ISB) serves as a classic approach for handling postoperative pain in the shoulder joint. Compared with general anesthesia alone, it can significantly decrease the length of hospital stay and alleviate postoperative pain scores [3]. Nevertheless, ISB is associated with numerous side effects, among which the relatively high incidence of phrenic nerve block (hemidiaphragmatic paralysis, HDP) has drawn the most attention. Patients with HDP might encounter about 30% reduction in the spirometric measurements of pulmonary function. A portion of them will state that they have mild dyspnea or changed respiratory sensations [4]. Although researchers have exerted considerable efforts, the incidence of HDP still hasn't been decreased to less than 20% [5].

Recently, several studies have revealed that during arthroscopic shoulder surgeries, the costoclavicular brachial plexus–cervical plexus blocks (CCB–CPBs) can also yield an analgesic effect comparable to that of the ISB. Meanwhile, the incidence of HDP associated with CCB–CPBs is lower than that of ISB, thus rendering it a potentially safer analgesic approach [6,7]. Subsequently, a cadaveric anatomical study uncovered that CCB can block the axillary nerve and the suprascapular nerve, which are the two major nerve branches responsible for innervating the sensation of the shoulder joint [8].

Consequently, the current evidence suggests that CCB-CPBs are capable of providing postoperative analgesia for arthroscopic shoulder surgeries. Nevertheless, an excessive volume of local anesthetic may still give rise to HDP [9]; conversely, an insufficient volume of local anesthetic might lead to inadequate analgesia. Hence, exploring the optimal dose of CCB–CPBs, which can offer a satisfactory

postoperative analgesic effect while averting HDP, represents an urgent issue that demands to be addressed in current clinical practice.

The aim of this study is to investigate the optimal dose of ropivacaine required for postoperative analgesia in patients undergoing arthroscopic shoulder surgeries under general anesthesia combined with CCB–CPBs. More precisely, this will be accomplished by identifying the dose that is effective for 50% of the cases ($ED_{50}$) and the dose that is effective for 95% of the cases ($ED_{95}$) of ropivacaine for the aforementioned objective.

## Methods

### Design

This trial will be a prospective, single-arm, double-blind dose finding study. The Ethics Committee of Ningbo No. 6 Hospital in Zhejiang Province, China, granted approval to the protocol of this study on July 29, 2024 (Approval No. 2024-67L). Subsequently, this study was registered at http://www.chictr.org.cn (Registration No. ChiCTR2400090292; Registration Date: September 26, 2024). In strict adherence to the Helsinki Declaration, we will obtain the informed written consent of all participants. This report will be developed in accordance with the principles laid out in Standard Protocol Items: Patient enrolment, interventions and assessments are presented in Fig 1. Recommendations for Interventional Trials (SPIRIT). Fig 2 shows the design of the study.

| | Enrolment | Allocation | Post-allocation | Follow-up |
|---|---|---|---|---|
| TIMEPOINT | *1 day before surgery* | surgery day | before surgery day | *PACU* |
| **ENROLMENT:** | | | | |
| Eligibility screen | X | | | |
| Informed consent | X | | | |
| Randomization | | X | | |
| Allocation | | X | | |
| **INTERVENTIONS:** | | | | |
| Nerve blocks | | | X | |
| **ASSESSMENTS:** | | | | |
| Numerical Rating Scale | | | | X |
| Diaphragmatic excursion measurement | | | X | X |
| Consumption of fentanyl | | | | X |
| Adverse events | | | | X |

**Fig 1. Schedule of enrolment, interventions and assessments for the trial.**

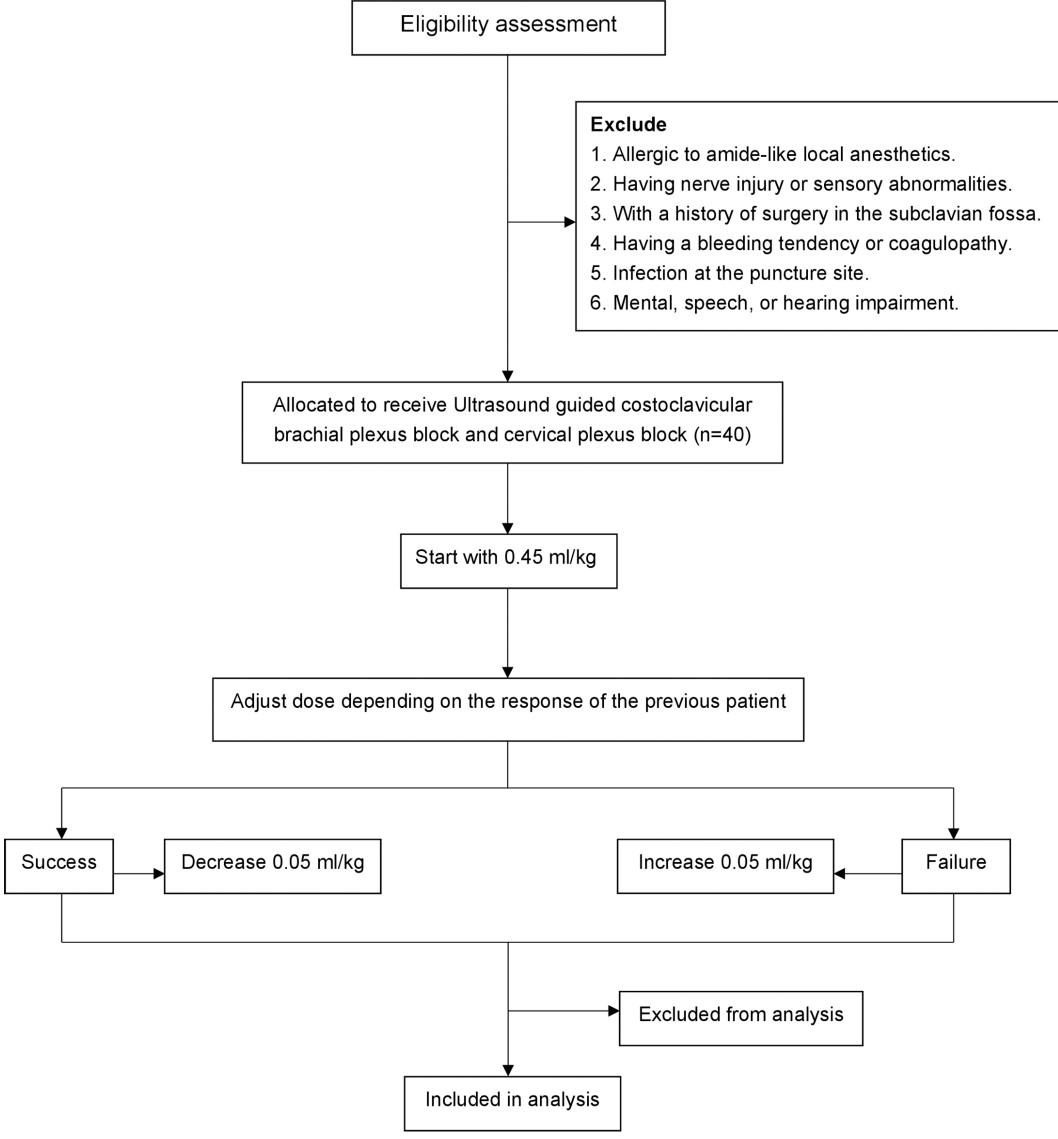

**Fig 2. Flow chart of patient recruitment.**

## Status and timeline

**Participant recruitment**: Participant recruitment commenced on October 8, 2024, and is currently underway. We antici-pate that the recruitment will be completed on October 8, 2025. This estimation is based on our recruitment plan and the current progress rate.

**Data collection**: The data collection will be completed simultaneously with the participant recruitment. As soon as the last participant is recruited on October 8, 2025, the data collection process will also reach its end. This is because our data collection protocol is designed to collect data from participants immediately upon their enrollment, and all necessary data can be obtained during the recruitment period.

**Results**: After the completion of data collection, we will start the data analysis process. Given the complexity of the data and the statistical methods we need to apply, we expect to have the results available by November 8, 2025.

## Subjects and Setting

Our intention will be to enroll 40 patients who will be scheduled to undergo arthroscopic shoulder surgeries under anesthesia that combines general anesthesia with CCB-CPBs at Ningbo No. 6 Hospital located in Ningbo, Zhejiang Province, China.

The inclusion criteria will be as follows:

1. American Society of Anesthesiologists (ASA) physical status I-III.

2. Age ranging from 18 to 75 years.

3. Body mass index within the range of 18–30 kg/m².

The exclusion criteria will be as follows:

1. Allergic to amide-like local anesthetics.

2. Having nerve injury or sensory abnormalities.

3. With a history of surgery in the subclavian fossa.

4. Having a bleeding tendency or coagulopathy.

5. Infection at the puncture site.

6. Mental, speech, or hearing impairment.

## Study protocol

No premedication will be given to the patients, and they will be asked to observe an eight-hour fasting period. Once they enter the operating room, we will establish a peripheral intravenous access in the non-operative upper limb using a 16-gauge catheter for fluid administration. A rapid peripheral intravenous preload using Ringer's lactate solution will be commenced at a rate of 10 ml/kg, and then the rate will be kept at a low rate throughout the entire procedure. Standard non-invasive monitoring methodologies, including non-invasive blood pressure measurement, pulse oximetry, and electrocardiography, will be implemented.

### Baseline diaphragmatic excursion measurement

The patient will be positioned in a semi-sitting position with the head raised approximately 30 degrees. Via the anterior subcostal route, either the liver or the spleen will serve as an acoustic window. The excursion of the hemidiaphragm will be measured by utilizing the SONOSITE SII® (FUJIFILM SonoSite, Inc., Bothell, USA) M-mode which will be equipped with a low-frequency (1–5 MHz) ultrasound probe. Stable waveforms will be documented during deep breathing, and the corresponding values will be recorded.

### Ultrasound guided costoclavicular brachial plexus block

The patient will lie flat on a horizontal operating table, with a soft cushion placed under the shoulder on the surgical side. The SONOSITE SII equipped with a high-frequency (6–13 MHz) linear array ultrasound probe will be utilized and positioned parallel to the inferior border of the clavicle on the affected side. The patient's head will be turned to the non-surgical side, and the affected limb will be moderately abducted to obtain the best view of the infraclavicular costoclavicular space. The ideal ultrasound view will be that the three trunks of the brachial plexus are arranged in sequence on the lateral side of the axillary artery. The anesthetist will adjust the direction and position of the ultrasound probe. After obtaining the best view, the intended puncture site will be disinfected under strict aseptic precautions. A 30-ml syringe

containing 30 ml of 0.5% ropivacaine (Naropin; AstraZeneca Co., Ltd.; 10 mg/ml) prepared prior to anesthesia will be used. The puncture site will be infiltrated and anesthetized with 2–3 ml of 1% lidocaine for skin anesthesia. Using the in-plane technique, a 21G 70mm disposable monopolar nerve block needle (Hakko Co., Ltd., Nagano, Japan) will be inserted from the outside to the inside. Considering that the axillary nerve, which innervates the shoulder, branches off from the posterior cord, we will position the injection site of the local anesthetic at the posterior cord.

## Weight-based allocation of local anesthetics

The volume of the local anesthetic will be ascertained by employing the small-sample Up-and-Down study design, which was described by Kewlani et al. [10]. Eligible subjects will be enrolled successively. The researcher (MZ) will set the initial volume at 0.45 ml/kg and adjust it in increments or decrements of 0.05 ml/kg, with the adjustment being decided by the response of the previous subject.

The anesthesiologist (LY) will be unaware of the volume of ropivacaine and will be responsible for the intraoperative and postoperative management. If the postoperative Numerical Rating Scale (NRS) score of the subject is ≤ 3 points, it will be defined as a successful block. LY will report the result to MZ, and MZ will prepare a ropivacaine volume one level lower when the next subject is enrolled. If the postoperative NRS score of the subject is > 3 points, it will be defined as a block failure. LY will report the result to MZ, and MZ will prepare a ropivacaine volume one level higher when the next subject is enrolled.

## Ultrasound cervical plexus block

Aiming to achieve anesthesia for the cape of the shoulder, considering that this area will be innervated by the supraclavicular nerves (C3 - C4), we will provide each subject with a superficial cervical plexus block guided by ultrasound [11]. As was reported previously, a volume of ten milliliters of 0.5% ropivacaine will be injected into the area between the sternocleidomastoid and scalene muscles at the fourth level of the cervical spine [7].

## General anesthesia

Upon completion of the nerve block procedure, all patients will be administered general anesthesia via a laryngeal mask airway. The induction agents will consist of intravenous propofol (at a dosage of 2–3 mg/kg) and atropine (in a dosage range of 0–0.5 mg). 1.5–3% sevoflurane will be utilized to maintain the anesthesia. After the induction of anesthesia, the assisted ventilation mode will be initially employed to assist patients with breathing.

When a skin incision will be made, the ventilation mode will be adjusted according to the patients' responses to the skin incision. For some patients, the pain stimulation during the skin incision will trigger spontaneous breathing, and we will promptly switch the ventilation mode to "spontaneous". If there is no response during the skin incision, the assisted ventilation mode will be maintained until the patients demonstrate sufficient spontaneous breathing, at which point we will change the ventilation mode once again.

When a skin incision will be made, the heart rate or blood pressure will be permitted to rise by 20% of the preoperative value, and the sevoflurane concentration will be adjusted to 3%. In case the heart rate and blood pressure do not decline within 15 minutes, intravenous boluses of 50 μg fentanyl will be administered, and the procedure will be repeated if considered necessary.

After the completion of surgery, each patient will be transferred to the post-anesthesia care unit (PACU). After the patients have been extubated and before leaving the PACU, we will score them based on their reported arm or shoulder discomfort, utilizing a NRS score in which 0 denotes no pain and 10 signifies the most intense anguish imaginable. In case the patient's NRS score is higher than 3, an intravenous bolus of 50 μg of fentanyl will be given to them, and this administration will be repeated as necessary.

## Measurements

The primary outcome is the patients' NRS score before leaving the PACU.

The secondary outcomes are as follows:

1. The ipsilateral diaphragmatic excursion.

2. The occurrence of complications, and the consumption of fentanyl.

Prior to the patients' departure from the PACU, we will measure the diaphragmatic excursion by employing the same methodology that will be utilized prior to the surgery. Complete HDP will be characterized as a 75–100% decrease in diaphragmatic excursion or the emergence of paradoxical movement. Partial HDP will be characterized as a 25–75% decrease in diaphragmatic excursion.

During the first thirty minutes subsequent to the induction of general anesthesia, the systolic blood pressure will be measured every three minutes; subsequently, it will be measured every five minutes. Hypotension will be regarded as a decrease in systolic blood pressure by more than 20% compared to the preoperative level. For patients with hypotension, 8 µg of norepinephrine could be intravenously injected once, and then this process will be continued until the blood pressure returns to 80% of its preoperative systolic level. The total amount of norepinephrine that will be administered will be recorded.

## Statistical considerations

Out of ethical considerations, in clinical trials, we will usually aim to calculate the most appropriate dose of a certain drug using the smallest possible sample size. Therefore, the small-sample Up-and-Down sequential method was selected based on its efficiency in estimating dose-response quantiles (e.g., $ED_{50}$, $ED_{95}$) with minimal sample sizes [12]. This adaptive design adjusts the local anesthetic volume for each subsequent patient based on the previous participant's response (success: NRS ≤ 3; failure: NRS > 3), ensuring rapid convergence toward the target dose range while minimizing exposure to subtherapeutic or toxic doses.

Sample size determination was guided by:

1. Primary Assumption: A standard deviation (SD) of 0.137 ml/kg and a standard error of the mean (SEM) of 0.043 ml/kg, derived from a pre-pilot study (n = 10).

2. Formula: The Dixon & Massey equation: $n = 2 \times \left( \frac{SD}{SEM} \right)^2$

3. Adjustment: To enhance precision for $ED_{95}$ estimation and align with simulation studies, the sample size was increased to 40 [13].

The $ED_{50}$ volume will be calculated via the Up-and-Down sequential method. The midpoint associated with a crossover (specifically, a shift from a successful block to an unsuccessful one) will be utilized for the calculation of the $ED_{50}$ volume. The Centered Isotonic Regression cran package of R software will be employed to calculate the $ED_{95}$ volume.

Continuous data will be subjected to normality testing using the Shapiro–Wilk test. For the data that will be normally distributed, they will be reported as the mean (SD), and differences will be evaluated by means of Student's t-test. In the case of non-normally distributed data, the Mann–Whitney U test will be employed for assessment. Fisher's exact test or the chi-square ($\chi^2$) test will be utilized to analyze the categorical data.

Statistical analyses will be conducted on the appropriate data using R software version 3.4.4, the SPSS 25.0 for Windows statistical package (SPSS, Inc., Chicago, IL). A P value less than 0.05 will be regarded as statistically significant.

### Data collection and management

Patient data will be collected and managed through ResMan (Research Manager, http://www.medresman.org.cn), a secure web-based clinical research data management platform certified by the Chinese Clinical Trial Registry. ResMan ensures data integrity through the following features:

• Role-Based Access Control: Only authorized personnel can access or modify data, with permissions tailored to study roles (e.g., investigators, auditors).

• Audit Trails: All data entries and modifications are automatically timestamped and logged, enabling full traceability.

• Real-Time Validation: Built-in logic checks prevent inconsistent or out-of-range entries (e.g., NRS scores exceeding 10).

To further minimize errors, two independent researchers will perform cross-verification of all data entries. Discrepancies will be resolved through consensus or adjudication by a third senior investigator.

The patient data will be collected and stored by the research team. Upon the completion of the study, the final data along with the informed consent forms of the subjects will be submitted to the relevant departments of Ningbo No. 6 Hospital in Zhejiang Province, China for archival storage. The detailed research particulars will be archived in the China Medical Research Registration and Filing Information System accessible at www.medicalresearch.org.cn.

## Discussion

Our research protocol seems to bear a strong resemblance to that of Kewlani et al study [10]. However, the main outcomes of the two studies differ completely. The optimal dose calculated by Kewlani et al was intended for forearm and hand surgeries, which also represent the common indications for most brachial plexus blocks. Nevertheless, our objective is to provide analgesia in shoulder surgeries. This is an effect that not all brachial plexus blocks can accomplish. It is extremely challenging to achieve a perfect analgesic effect in shoulder surgeries while simultaneously avoiding the occurrence of HDP. Over the years, numerous researchers have exerted diverse efforts in an attempt to avoid HDP, yet HDP has never been entirely avoidable [5].

Avoiding HDP is of great clinical significance. Given that HDP events are binary, all-or-nothing phenomena, implying that even the slightest risk may pose a substantial hidden danger [14]. For instance, when an anesthesiologist applies a certain brachial plexus block to a patient with severe pulmonary lesions, although the incidence of HDP with this technique is quite low, once it occurs, the consequence is that the patient will suffer from dyspnea and require emergency intubation. And the costs associated with this are extremely high. Therefore, avoiding HDP is of vital importance in clinical practice.

To date, only the CCB-CPBs have been reported for the first time to achieve successful anesthesia of the shoulder while simultaneously avoiding HDP [7]. All in all, this finding is of considerable significance and points the way for the development of shoulder analgesia techniques free of HDP. This study endeavors to verify their research findings and, concurrently, seek the most appropriate dose. Consequently, the purpose and significance of this study are entirely distinct from those of Kewlani et al study.

A previous study compared the analgesic effects of 20 ml and 40 ml of CCB in shoulder surgeries, and the findings demonstrated that there was no difference in the analgesic effects between the two [15]. The outcome of this study might prompt readers to question our research design. Specifically, given that a volume difference of 20 ml failed to enhance the analgesic effect, one might wonder whether the incremental increase of 0.05 ml/kg in our study would bring about any changes to the analgesia.

Firstly, we commend the research of Yumin Jo et al on the impact of volume changes on the supraclavicular spread of CCB and its analgesic effect. In particular, their direct observation using ultrasound to determine whether the local anesthetic spreads to the suprascapular nerve (SSN) after CCB is highly valuable. It provided the second direct evidence, aside from anatomical studies, that CCB can spread to the SSN. Encouragingly, their research results indicated that 20 ml

of CCB blocked the SSN in 56% of the subjects. However, disappointingly, 40 ml of CCB only blocked the SSN in 60% of the subjects. Such results regarding SSN block were also corroborated by the postoperative analgesic effects, meaning that there was no significant difference between the two groups. In the study of Quehua Luo et al, an 87% SSN block rate was achieved among the subjects with 20 ml of CCB. It is worth highlighting that, in the control group with ISB, the SSN block rate was merely 91% [16]. In another study evaluating the effect of the brachial plexus block involving the upper trunk, although the authors claimed that all the nerves including the SSN achieved sensory and motor block, there were still 6.6% of the subjects who experienced incision pain and 3% of the subjects who had tourniquet reactions respectively [17].

The results of the aforementioned studies imply that even in the case of ISB, the SSN block rate remains below 100%. The probable reason lies in the fact that the SSN branches off from the upper trunk prematurely and might not be blocked even at the C5–C6 level. We posit that the SSN block rate for 20 ml of CCB could be 87%. The rationale behind this is that the sample size of this study is relatively large (n = 106), and the evaluation method is fairly standardized [16]. There could be certain errors when employing ultrasound for direct observation. Hence, an 87% SSN block rate is reasonable for 20 ml of CCB, considering that even if 40 ml of CCB can spread to the C5–C6 level, the SSN block rate will be at most 91%.

Consequently, the objective of our study might be to determine the most appropriate volume for CCB to effectively block the SSN. Based on the available data, the 90% effective dose ($ED_{90}$) might be approximately 40 ml. Nevertheless, given that we have incorporated the cervical plexus block in addition to CCB, as reported in the literature, the local anesthetic used in the cervical plexus block might permeate through the prevertebral fascia and impact the phrenic nerve as well as the C5 nerve [18]. Hence, it is anticipated that the $ED_{90}$ might decrease.

### Strengths and limitations of this study

- This study will determine the optimal volume of 0.5% ropivacaine for postoperative analgesia for patients undergoing arthroscopic shoulder surgeries under general anesthesia with costoclavicular brachial plexus–cervical plexus blocks.

- The optimal volume of costoclavicular brachial plexus–cervical plexus blocks can provide effective analgesia while minimizing the occurrence of hemidiaphragmatic paralysis as much as possible.

- This is a single-centre, single-arm trial that does not have a control arm.

- The sample size of the study is small.

### Supporting information

**S1 Checklist.  SPIRIT Checklist for study protocol adherence.**
(DOCX)

**S1 Chinese Protocol.  Chinese version of the original study protocol.**
(DOCX)

**S2 English Protocol.  English version of the original study protocol.**
(DOCX)

### Author contributions

**Conceptualization:** Liyong Yuan, Zhong Mei, Zhimin Sheng, Xiaolu Huang, Miao Zhu.

**Data curation:** Jianlin Wang, Miao Zhu.

**Formal analysis:** Zhimin Sheng, Miao Zhu.

**Funding acquisition:** Miao Zhu.

**Investigation:** Miao Zhu.

**Methodology:** Jianlin Wang, Liyong Yuan, Zhong Mei, Zhimin Sheng, Xiaolu Huang, Miao Zhu.

**Project administration:** Jianlin Wang, Liyong Yuan, Xiaolu Huang, Miao Zhu.

**Resources:** Liyong Yuan.

**Software:** Xiaolu Huang.

**Supervision:** Liyong Yuan, Zhong Mei, Zhimin Sheng.

**Validation:** Jianlin Wang, Zhong Mei, Xiaolu Huang.

**Visualization:** Xiaolu Huang.

**Writing – original draft:** Jianlin Wang, Miao Zhu.

**Writing – review & editing:** Jianlin Wang, Liyong Yuan, Zhong Mei, Zhimin Sheng, Xiaolu Huang, Miao Zhu.

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
