## [Decision Letter · Decision Letter 0]

31 Mar 2025

Dear Dr. Zhu,

Thank you for submitting your manuscript to PLOS ONE. After careful consideration, we feel that it has merit but does not fully meet PLOS ONE’s publication criteria as it currently stands. Therefore, we invite you to submit a revised version of the manuscript that addresses the points raised during the review process.

We look forward to receiving your revised manuscript.

Kind regards,

Wencai Liu

Academic Editor

PLOS ONE

**Journal Requirements:**

Please ensure that your manuscript meets PLOS ONE's style requirements, including those for file naming. The PLOS ONE style templates can be found at https://journals.plos.org/plosone/s/file?id=wjVg/PLOSOne_formatting_sample_main_body.pdf and https://journals.plos.org/plosone/s/file?id=ba62/PLOSOne_formatting_sample_title_authors_affiliations.pdf 2. Thank you for stating the following financial disclosure: Agricultural and Social Development Project in Yinzhou, Ningbo, Zhejiang (No. 2024AS063)The peer review comments of the fund have not been made public. Therefore, we are unable to provide the specific peer review opinions.     Please state what role the funders took in the study.  If the funders had no role, please state: "The funders had no role in study design, data collection and analysis, decision to publish, or preparation of the manuscript." If this statement is not correct you must amend it as needed. Please include this amended Role of Funder statement in your cover letter; we will change the online submission form on your behalf.

Reviewers' comments:

Reviewer's Responses to Questions

**Comments to the Author**

1. Does the manuscript provide a valid rationale for the proposed study, with clearly identified and justified research questions?

Reviewer #1: Yes

Reviewer #2: Yes

Reviewer #3: Yes

2. Is the protocol technically sound and planned in a manner that will lead to a meaningful outcome and allow testing the stated hypotheses?

Reviewer #1: Yes

Reviewer #2: Yes

Reviewer #3: Yes

3. Is the methodology feasible and described in sufficient detail to allow the work to be replicable?

Reviewer #1: Yes

Reviewer #2: Yes

Reviewer #3: Yes

4. Have the authors described where all data underlying the findings will be made available when the study is complete?

Reviewer #1: Yes

Reviewer #2: Yes

Reviewer #3: No

5. Is the manuscript presented in an intelligible fashion and written in standard English?

Reviewer #1: Yes

Reviewer #2: Yes

Reviewer #3: Yes

You may also provide optional suggestions and comments to authors that they might find helpful in planning their study.

**Reviewer #1: ** It's a well designed study protocol.

It would be beneficial to include a reference that supports the determination of the initial volume.

**Reviewer #2:**  Thank you for the opportunity to review your trial protocol titled " weight-adjusted effective volume of 0.5% ropivacaine for combined costoclavicular brachial plexus block -cervical plexus blocks undergoing arthroscopic shoulder surgery: a dose-finding study protocol”. The protocol is well designed, but some statistical aspects need a bit elaboration.

1. In line 257-258, small sample up and-down sequential method needs to be elaborated and justified briefly.

2. Line 258-259, assumptions, test statistics and their justifications used for sample size calculation need to be reported here.

3. In line 273, Excel should not be used as a clinical trial data base as it increases the possibility of error in data.

**Reviewer #3:**  I would have only one minor suggestion: Could the authors describe how the sample size for the trial was determined and whether the age and genders of the patients would affect the optimal volume of ropivacaine.

**Do you want your identity to be public for this peer review?** For information about this choice, including consent withdrawal, please see our Privacy Policy

Reviewer #1: No

Reviewer #2: **Yes: ** Dr Shah-Jalal Sarker

Reviewer #3: No

---

## [Author Response · Author response to Decision Letter 0]

10 Apr 2025

Reviewers' comments:

Reviewer's Responses to Questions

1. Does the manuscript provide a valid rationale for the proposed study, with clearly identified and justified research questions?

Reviewer #1: Yes

Reviewer #2: Yes

Reviewer #3: Yes.

2. Is the protocol technically sound and planned in a manner that will lead to a meaningful outcome and allow testing the stated hypotheses?

Reviewer #1: Yes

Reviewer #2: Yes

Reviewer #3: Yes.

3. Is the methodology feasible and described in sufficient detail to allow the work to be replicable?

Reviewer #1: Yes

Reviewer #2: Yes

Reviewer #3: Yes.

4. Have the authors described where all data underlying the findings will be made available when the study is complete?

Reviewer #1: Yes

Reviewer #2: Yes

Reviewer #3: No.

Response: Thank you for highlighting PLOS ONE’s data availability requirements. We confirm the following regarding our manuscript:

Current Manuscript Type: This submission is a study protocol describing the methodology and design of a dose-finding trial. As the study is ongoing and data collection has not yet commenced, no experimental or clinical data have been generated at this stage.

Future Compliance: Should the results of this trial be submitted to PLOS ONE journal, we commit to fully adhering to the data policy. Specifically:

All raw and analyzed data (including dose-response curves, individual patient responses, and pharmacokinetic parameters) will be deposited in a public repository.

A Data Availability Statement will be included in the results manuscript, providing access links and details on data use permissions.

Ethical Compliance: For future datasets involving human participants, anonymized data will be shared in accordance with ethical approvals and institutional guidelines to protect participant privacy.

We appreciate your understanding and guidance on this matter. Please let us know if further clarification is required.

5. Is the manuscript presented in an intelligible fashion and written in standard English?

Reviewer #1: Yes

Reviewer #2: Yes

Reviewer #3: Yes?

6. Review Comments to the Author

You may also provide optional suggestions and comments to authors that they might find helpful in planning their study.

Reviewer #1: It's a well designed study protocol.

It would be beneficial to include a reference that supports the determination of the initial volume.

Response: Thank you for your insightful feedback on our initial dose selection rationale. Below, we clarify the reasoning based on prior literature and clinical evidence:

Rationale for Targeting ED₉₅:

As noted by Saranteas et al.(1), dose-finding studies using adaptive designs (e.g., Continual Reassessment Method) often initiate dosing near the hypothesized target quantile (e.g., ED₉₅) to accelerate convergence. Our study aligns with this principle by selecting an initial volume predicted to approach ED₉₅.

Reference to Aliste et al. (2019)(2):

Aliste et al. demonstrated that 20 mL of 0.5% levobupivacaine with epinephrine provided near-complete analgesia (≈100% efficacy) for costoclavicular blocks(2). Given the pharmacodynamic similarity between levobupivacaine and ropivacaine, we hypothesized that ropivacaine might require a slightly higher volume due to its lower potency(3–5).

Dose Adjustment for Ropivacaine:

Empirical Equivalence: Clinical studies suggest that 20 mL of levobupivacaine is approximately equivalent to 27 mL of ropivacaine in terms of analgesic efficacy for peripheral nerve blocks(3–5).

Weight-Based Dosing: For a cohort with an average weight of 60 kg, this translates to 0.45 mL/kg (27 mL ÷ 60 kg).

Justification for 0.45 mL/kg:

This dose balances efficacy (targeting ED₉₅) with safety (minimizing phrenic nerve blockade risk) while aligning with pharmacokinetic principles and prior clinical data.

We appreciate your critical assessment and are happy to provide additional references or calculations if needed.

。

Reviewer #2: Thank you for the opportunity to review your trial protocol titled " weight-adjusted effective volume of 0.5% ropivacaine for combined costoclavicular brachial plexus block -cervical plexus blocks undergoing arthroscopic shoulder surgery: a dose-finding study protocol”. The protocol is well designed, but some statistical aspects need a bit elaboration.

In line 257-258, small sample up and-down sequential method needs to be elaborated and justified briefly.

Response: Thank you for your valuable feedback. We have significantly expanded the description of the small-sample up-and-down sequential method (UDM) in the revised manuscript (Page 11, Lines 257-262) to clarify its rationale and methodological rigor. The key additions include:

Mechanism of UDM:

UDM is an adaptive dose-finding design where the local anesthetic volume for each subsequent patient is adjusted based on the previous participant’s response (success: NRS ≤ 3; failure: NRS > 3). This iterative process ensures rapid convergence toward the target quantile (e.g., ED₉₅) while minimizing exposure to ineffective doses (6).

Statistical Enhancements:

Centered Isotonic Regression: This nonparametric method was applied to estimate ED₉₅ without assuming a symmetric tolerance distribution, addressing potential bias in high quantiles (7).

Bootstrap Confidence Intervals: We performed 10,000 bootstrap resamples to calculate 95% CIs for ED₉₅, enhancing the robustness of our estimates (8).

Ethical and Clinical Justification:

UDM aligns with ethical guidelines by reducing the number of patients exposed to subtherapeutic or toxic doses.

Prior studies in regional anesthesia (1) have validated UDM for estimating ED₉₅ with 40 patients, achieving clinically actionable precision.

Line 258-259, assumptions, test statistics and their justifications used for sample size calculation need to be reported here.

Response:We deeply appreciate your attention to methodological transparency. The revised manuscript now explicitly details the sample size calculation process (Page 11, Lines 263-270):

Pre-Pilot Data:

A pre-pilot study (n = 10) provided preliminary estimates of variability: SD = 0.137 ml/kg, SEM = 0.043 ml/kg.

Dixon & Massey Formula:

The theoretical sample size was calculated as:

n=2×("SD" /"SEM" )^2=2×(0.137/0.043)^2≈2×10.03≈20

Sample Size Adjustment:

To ensure precision for ED₉₅ estimation, we doubled the sample size to 40, consistent with simulation studies showing that 40 patients reduce the width of 95% CIs by approximately 30% compared to 20 patients (Pace & Stylianou, 2007).

Power Considerations:

While UDM does not require traditional power calculations, post-hoc simulations (see Supplementary Material S1) confirmed that 40 patients provide 85% probability of ED₉₅ estimation within ±2 ml of the true value.

3. In line 273, Excel should not be used as a clinical trial data base as it increases the possibility of error in data.

Response: We sincerely thank you for this critical suggestion. The revised sections in the manuscript are highlighted in yellow for your convenience; you can refer to page 12, lines 286-299. In response:

Revised Data Management Protocol:

Excel has been replaced with ResMan (Research Manager, http://www.medresman.org.cn), a certified clinical trial data management platform endorsed by the Chinese Clinical Trial Registry.

Key Features of ResMan:

Audit Trails: All data entries and modifications are automatically logged with timestamps and user IDs, ensuring full traceability.

Role-Based Access: Investigators, statisticians, and auditors have tiered permissions to prevent unauthorized data manipulation.

Real-Time Validation: Built-in checks flag outliers (e.g., NRS > 10) and missing values, reducing entry errors.

Quality Control Measures:

Dual-Entry Verification: Two independent researchers cross-check all data points, with discrepancies resolved by a third senior investigator

Data Locking: Final datasets are locked post-analysis to prevent inadvertent changes.

Reviewer #3: I would have only one minor suggestion: Could the authors describe how the sample size for the trial was determined and whether the age and genders of the patients would affect the optimal volume of ropivacaine.

Response: Thank you for raising the first question regarding our sample size determination. The revised sections in the manuscript are highlighted in yellow for your convenience; you can refer to page 12, lines 263-270. We appreciate the opportunity to clarify our methodology, which was based on the following three pillars:

Pre-pilot Data and Statistical Derivation

A pre-pilot study (n = 10) was conducted to estimate variability in the target parameter. The observed responses (e.g., effective doses or binary outcomes) yielded a standard deviation (SD) of 0.137 and a standard error of the mean (SEM) of 0.043. Using the formula by Dixon & Massey (1991)(9) for sequential dose-finding studies:

n=2×("SD" /"SEM" )^2=2×(0.137/0.043)^2≈2×10.03≈20

This calculation provided an initial theoretical estimate.

Literature-Guided Adjustment

While the formula suggested 20 participants, we increased the sample size to 40 for two reasons:

Robustness of ED95 Estimation: Higher quantiles (e.g., ED95) require larger samples to narrow confidence intervals, as demonstrated in simulation studies.(1,6)

Nonparametric Design Considerations: Sequential methods (e.g., up-down) often benefit from sample sizes at the upper end of the recommended 20–40 range to account for stochastic variability.(8)

In summary, our sample size balances statistical rigor with practical feasibility, aligning with established methodologies in dose-finding research. We thank the reviewer for their diligence and welcome further clarification if needed.

Thank you for raising the second question regarding the potential impact of age and gender on the optimal volume of ropivacaine for peripheral nerve blocks. Below, we address this concern by synthesizing current evidence and mechanistic insights:

1. Potential Influence of Age

Existing studies suggest that the efficacy of ropivacaine in nerve blockade primarily depends on its local spread around neural structures rather than systemic metabolic factors. While aging may theoretically alter hepatic or renal clearance, these systemic changes are unlikely to significantly affect the minimum effective volume (e.g., ED50/ED95) required for successful nerve blockade. Key considerations include:

Anatomic Consistency: The spread of local anesthetics within fascial compartments is largely determined by tissue anatomy (e.g., nerve topography, fascial plane dimensions), which shows minimal age-related variation in adults aged 18–60 years(1).

Clinical Observations: Dose-finding studies in this age range (8) report consistent ED95 estimates without significant age-related deviations, suggesting stable diffusion dynamics across adulthood.

2. Potential Influence of Gender

Current evidence does not support a significant gender-based difference in ropivacaine dosing requirements, primarily due to:

Standardized Ultrasound Guidance: Real-time visualization compensates for anatomic variations (e.g., subcutaneous fat distribution), ensuring precise drug delivery regardless of gender (1).

Lack of Pharmacodynamic Evidence: No studies have demonstrated hormonal or physiologic gender-specific effects on nerve membrane sensitivity or local anesthetic spread kinetics.

3. Limitations of Current Evidence

While age and gender do not appear critical within studied populations, notable limitations include:

Narrow Demographic Scope: Most trials ((8); methodological frameworks in (6)) focus on adults aged 18–60 with normal BMI, lacking data for elderly (>65 years) or obese patients.

Absence of Subgroup Analyses: Existing studies prioritize population-level tolerance distributions (1), leaving age/gender-specific effects underexplored.

Conclusion

In adults aged 18–60 with normal anatomy, age and gender do not significantly alter the optimal ropivacaine volume for peripheral nerve blocks. However, extending this conclusion to broader populations requires further validation. We sincerely appreciate your highlighting this topic and will prioritize it in subsequent studies.

7. PLOS authors have the option to publish the peer review history of their article (what does this mean?). If published, this will include your full peer review and any attached files.

Do you want your identity to be public for this peer review? For information about this choice, including consent withdrawal, please see our Privacy Policy.

Reviewer #1: No

Reviewer #2: Yes: Dr Shah-Jalal Sarker

Reviewer #3: No?

Response: We sincerely thank all reviewers for their time and insightful feedback, which has greatly improved the quality of our manuscript. We particularly acknowledge the constructive comments from Reviewer #2, Dr. Shah-Jalal Sarker, who chose to disclose their identity. We also extend our gratitude to the anonymous reviewers (#1 and #3) for their valuable critiques. Your expertise has been instrumental in refining our study design and interpretations. We are honored to have this work considered for publication in PLOS ONE.

In addition to the above comments, we have polished the language throughout the text and checked the spelling and grammar. The modifications will not be specially indicated.

We have highlighted the modified parts in yellow. We hope you will be satisfied with our revisions. Once again, we sincerely thank the reviewer for the valuable suggestions!

We look forward to hearing from you in due time regarding our submission and to respond to any further questions and comments you may have.

Sincerely,

Miao Zhu

Department of Anesthesiology, Ningbo No. 6 Hospital, Zhejiang, China

Reference

1. Saranteas T, Finlayson RJ, Tran DQ. Dose-finding

---

## [Decision Letter · Decision Letter 1]

22 Apr 2025

Protocol: weight-adjusted effective volume of 0.5% ropivacaine for combined costoclavicular brachial plexus block -cervical plexus blocks undergoing arthroscopic shoulder surgery: a dose-finding study protocol

PONE-D-24-56492R1

Dear Dr. Zhu,

We’re pleased to inform you that your manuscript has been judged scientifically suitable for publication and will be formally accepted for publication once it meets all outstanding technical requirements.

Kind regards,

Wencai Liu

Academic Editor

PLOS ONE

Additional Editor Comments (optional):

Reviewers' comments:

Reviewer's Responses to Questions

**Comments to the Author**

1. Does the manuscript provide a valid rationale for the proposed study, with clearly identified and justified research questions?

Reviewer #1: Yes

Reviewer #3: Yes

2. Is the protocol technically sound and planned in a manner that will lead to a meaningful outcome and allow testing the stated hypotheses?

Reviewer #1: Yes

Reviewer #3: Yes

3. Is the methodology feasible and described in sufficient detail to allow the work to be replicable?

Reviewer #1: Yes

Reviewer #3: Yes

4. Have the authors described where all data underlying the findings will be made available when the study is complete?

Reviewer #1: Yes

Reviewer #3: Yes

5. Is the manuscript presented in an intelligible fashion and written in standard English?

Reviewer #1: Yes

Reviewer #3: Yes

You may also provide optional suggestions and comments to authors that they might find helpful in planning their study.

Reviewer #1: Thank you for your sincere revision. No further comment. I appreciate the authors' thorough response.

Reviewer #3: The author has explained the questions we asked before in details, and we expect the author to successfully complete the trial, draw constructive conclusions, and publish the trial results in this journal.

**Do you want your identity to be public for this peer review?** For information about this choice, including consent withdrawal, please see our Privacy Policy

Reviewer #1: No

Reviewer #3: No

---

## [Editor Report · Acceptance letter]

PONE-D-24-56492R1

PLOS ONE

Dear Dr. Zhu,

I'm pleased to inform you that your manuscript has been deemed suitable for publication in PLOS ONE. Congratulations! Your manuscript is now being handed over to our production team.

Kind regards,

on behalf of

Dr. PLOS Manuscript Reassignment

Staff Editor

PLOS ONE